# Bootstrapping Task Spaces for Self-Improvement

**Minqi Jiang**                                                                          *msj@meta.com*
*Meta Superintelligence Labs*

**Andrei Lupu**
*Meta Superintelligence Labs & University of Oxford*

**Yoram Bachrach**
*Meta Superintelligence Labs*

**Reviewed on OpenReview:** *https://openreview.net/forum?id=k2VsgUxC6X*

## Abstract

Progress in many task domains emerges from repeated revisions to previous solution attempts. Training agents that can reliably self-improve over such sequences at inference-time is a natural target for reinforcement learning (RL), yet the naive approach assumes a fixed maximum iteration depth, which can be both costly and arbitrary. We present Exploratory Iteration (ExIt), a family of autocurriculum RL methods that directly exploits the recurrent structure of self-improvement tasks to train LLMs to perform multi-step self-improvement at inference-time while only training on the most informative single-step iterations. ExIt grows a task space by selectively sampling the most informative intermediate, partial histories encountered during an episode for continued iteration, treating these starting points as new self-iteration task instances to train a self-improvement policy. ExIt can further pair with explicit exploration mechanisms to sustain greater task diversity. Across several domains, encompassing competition math, multi-turn tool-use, and machine learning engineering, we demonstrate that ExIt strategies, starting from either a single or many task instances, can produce policies exhibiting strong inference-time self-improvement on held-out task instances, and the ability to iterate towards higher performance over a step budget extending beyond the average iteration depth encountered during training.

## 1 Introduction

When given more time to think, humans can often reason toward better solutions to difficult problems. Recently, reinforcement learning (RL) on task-specific rewards has been shown to induce similar reasoning capabilities in pretrained large language models (LLMs) (Zelikman et al., 2022; Jaech et al., 2024; Guo et al., 2025), whereby the model learns to output extended chains-of-thought (CoTs) exhibiting reasoning behaviors that can iteratively self-improve a solution, via tactics like correctness verification and backtracking. This approach has been applied in both *verifiable* domains, such as math and coding, where an oracle function determining the correctness of any given solution can be explicitly defined (Lambert et al., 2024), and more recently extended to *non-verifiable* domains like creative writing (Gurung & Lapata, 2025; Zhou et al., 2025). Training such *reasoning models* via RL can be challenging: Empirically, eliciting useful reasoning patterns is dependent on how likely it is for the base model to generate CoTs already exhibiting these patterns (Gandhi et al., 2025), and thus a supervised fine-tuning stage using carefully-curated reasoning traces may be necessary. As CoTs lengthen with multiple reasoning steps, response generation takes longer to complete, slowing down training and potentially exhausting context length and memory.

In this work, we consider a complementary approach to improving an LLM's capacity for self-improvement at inference time, based on training the LLM to perform $K$-step self-improvement. In this problem setting, the LLM is presented with an initial solution and has a budget of $K$ steps to reach the best-performing

solution, with each subsequent step starting from the response produced in the previous step, along with an optional feedback signal communicating the quality of that solution.

Naively, $K$-step self-improvement training may be performed by randomizing over the step budget $K$, with the reward for each step corresponding to either the absolute quality of the solution produced that step or the relative improvement with respect to the previous solution. However, this approach is both overly restrictive and costly, requiring commitment to a maximum self-improvement depth of $K$ beforehand, while also increasing inference costs by an additional factor of $(K+1)/2$ on average, when maintaining a fixed number of training task instances. Moreover, this approach does not extend naturally to the multi-turn task setting, where each turn may be allotted a budget of $K$-steps for self-improvement before arriving at a final response for that step. Lastly, RL fine-tuning based on verified rewards can reduce output diversity, making it harder to find effective improvements to previous solutions (Wu et al., 2025).

Our approach avoids these issues by training for $K$-step self-improvement using only single-step self-improvement transitions. The starting point is the output of a previous turn, selected via prioritized sampling using a score that quantifies the learning potential of reattempting that turn given its exact history. Each training task comprises the original instruction, the sampled turn's previous output, and (in multi-turn settings) the remaining turn history up to that point. By teaching the LLM to iterate on its own solutions, our method can be viewed as self-generated data augmentation over the task space. To maintain diversity during iteration, we study two exploration mechanisms: (1) a variant of $\epsilon$-greedy exploration where, with probability $\epsilon$, the task switches from self-improvement to *self-divergence*, aiming for a valid but substantially different approach; and (2) multiplicatively scaling the group-wise advantage of each rollout by its distance from the group centroid in an embedding space. Because any step endpoint can seed a new iteration task, step-level and task-level exploration act in the same task space, blurring the line between classic step-level exploration and task-level autocurricula. We refer to our approach as Exploratory Iteration (ExIt).

We demonstrate the effectiveness of ExIt variants across a variety of domains—including competition math, multi-turn tool-use, and machine learning engineering tasks based on previous Kaggle competitions—where policies trained via ExIt variants exhibit improved $K$-step self-improvement at inference-time, with $K$ extending beyond the typical improvement depth seen during training. Further, we observe that ExIt induces emergent autocurricula over various task complexity metrics while naturally increasing the diversity of the training task instances, leading to improved performance even before self-improvement.

## 2 Background

### 2.1 Self-improvement decision process

We consider the $K$-step self-improvement setting in a task space $\mathcal{M}$ defining a set of task instances $m$, each corresponding to a multi-turn POMDP with a timestep budget of $T_m$. For clarity and consistency with recent usage for LLM-based tasks, we refer to the timesteps of the underlying POMDP as *turns* and reserve *step* to refer to a self-iteration step, where the LLM generates a modified version of a previous response.

For a task instance $m$, at every turn $t$, for $1 \le t \le T_m$, the LLM $\pi_\theta$ with parameters $\theta$, receives the task prompt (also denoted as $m$ for convenience) and outputs a response containing a solution to the request $y_t$, followed by $k \ge 0$ additional steps of self-iteration. Each iteration step acts on the latest solution iterate to produce the next iterate, $y_t^k$, where we define $y_t^0 \equiv y_t$, so the improve step with index 0 corresponds to the initial response attempt for that turn. At each iteration, $\pi$ also observes the remaining history, $y_{<t}^K$, composed of the sequence of the last iterates per turn, up to the last fully-iterated turn $t-1$. Optionally, associated feedback signals $e_t^k$ and $e_{<t}^K$, e.g. execution traces in a coding task, can be provided to $\pi$ for each observed iterate. For convenience, let the history $\tau_t^k \equiv (y_t^k, y_{<t}^K, e_t^k, e_{<t}^K)$. Then,

$$y_t^{k+1} \sim \pi_\theta(\cdot|\tau_t^k, m). \tag{1}$$

Let the final history be $\tau_{T_m}^K \equiv \hat{\tau}_m$. Further, let $G$ be a function mapping $\hat{\tau}$ to a real scalar measuring the quality of the responses. In this work $G$ is the undiscounted sum of rewards $r_t$ at the end of each turn $t$ in $\hat{\tau}$. We then seek the optimal policy parameters maximizing the quality of the solution represented by $\hat{\tau}$,

$$\arg\max_{\theta} \mathbb{E}_{\substack{m \sim \mathcal{M} \\ \hat{\tau}_m \sim \pi_\theta}} \big[\, G(\hat{\tau}_m) \,\big]. \tag{2}$$

Let us define a *self-improvement task* as a tuple comprising an initial task $m$, a partial history $\tau_t^k$, and a self-improvement budget $K$, with the objective of maximizing the quality of the final solution iterate $\hat{y}_m$ reached via $K$ improvement steps per turn starting from $\tau_t^k$. We further define a *recurrent task* as a task for which the output of one instance can serve as the input parameter to define another instance. Self-improvement tasks are recurrent tasks.

## 2.2 Group-Relative Policy Optimization

We search for the optimal LLM parameters $\theta$ in Equation 2 using Group-Relative Policy Optimization (GRPO) (Shao et al., 2024). GRPO is an online policy-gradient algorithm that optimizes a clipped advantage objective based on that used in PPO. Rather than computing the baseline term using a learned value function as in PPO (Schulman et al., 2017), GRPO uses a group of $G$ Monte-Carlo rollouts, $\{o_i\}_{i=1}^G$ for each initial prompt $m$ to estimate this baseline. We use GRPO for its strong empirical performance and reduced resource requirements compared to PPO from dropping the learned value function, typically a second copy of the LLM. Let $\mathbf{r}$ be the vector of returns for each rollout, where $r_i$ is the return for rollout $o_i$. Within each group, the advantage $A_{i,t}$ for token index $t$ in rollout $o_i$ is

$$A_{i,t} = \frac{r_i - \mathrm{mean}(\mathbf{r})}{\mathrm{std}(\mathbf{r})}, \tag{3}$$

where the same advantage is assigned to every completion token index $t$. GRPO makes use of a KL regularization term constraining $\pi_\theta$ to a reference policy $\theta_{\mathrm{ref}}$, initialized with the initial parameters, and updated to the latest $\theta$ with a $1 - \alpha$ decay factor every $M$ training iterations. The full GRPO objective is then

$$\mathcal{J}(\theta) = \mathbb{E}_{\substack{m \sim \mathcal{M}, \\ \{o_i\}_{i=1}^G \sim \pi_{\mathrm{old}}(O|m)}} \frac{1}{G} \sum_{i=1}^{G} \frac{1}{|o_i|} \sum_{t=1}^{|o_i|} \left[ \min\big[\, \rho_{i,t}\, A_{i,t}\,,\, \mathrm{clip}(\rho_{i,t},\, 1-\varepsilon,\, 1+\varepsilon)\, A_{i,t}\big] \;-\; \beta\, D_{\mathrm{KL}}\big[\pi_\theta \,\|\, \pi_{\mathrm{ref}}\big] \right], \tag{4}$$

where $\pi_{old}$ is the set of parameters used to generate the rollouts, $\varepsilon > 0$ is the clip threshold, $\beta > 0$ is the KL coefficient, and the importance sampling coefficient for $o_i$ at token index $t$ is

$$\rho_{i,t} \;=\; \frac{\pi_\theta\big(o_{i,t} \mid m,\, o_{i,<t}\big)}{\pi_{\mathrm{old}}\big(o_{i,t} \mid m,\, o_{i,<t}\big)}. \tag{5}$$

# 3 Task diversity improves learnability

A key observation motivating EXIT is that the learnability of individual tasks can be improved by training on additional tasks with related structure. In Figure 1, we see DeepSeek-R1-Distill-Qwen-7B cannot directly learn the MLE-bench task random-acts-of-pizza, while simultaneously training on two additional MLE-bench tasks enables learning. Cross-task transfer is popularly exploited by methods like domain randomization (Tobin et al., 2017; Mehta et al., 2020), where a training distribution over varied task instances is sampled to improve task diversity, and curriculum learning methods, which actively sample the task space. The specific approach used for sampling task instances defines an exploration strategy over the task space. The recurrence in self-improvement tasks introduces a natural source of related tasks, as each solution iterate itself can serve as a task instance.

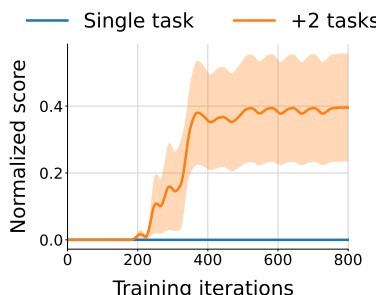

Figure 1: Simultaneously training with other tasks makes random-acts-of-pizza learnable.

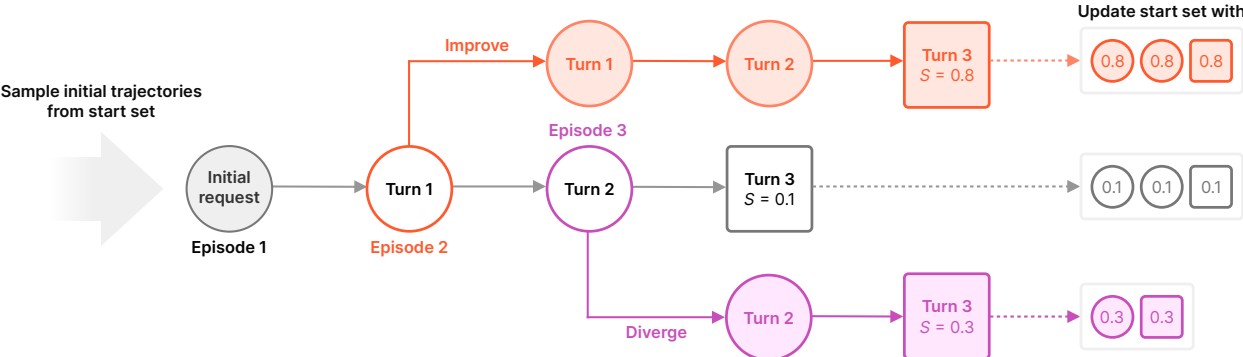

Figure 2: Overview of ExIt strategies. Each episode samples a new task (at turn 0) or selects a partial turn history from a previous episode as a starting point for self-iteration (either self-improvement or divergence). Partial histories are sampled by prioritizing those that led to higher GRPO group return variance.

Our approach, ExIt, exploits this fact by actively exploring across the history of solution iterates, both by actively sampling promising previous iterates for further iteration and by promoting iteration steps to output novel solutions. In this way, ExIt seeks to produce more robust self-improvement abilities via cross-task transfer across a potentially open-ended set of self-generated starting solutions.

## 4 Exploratory Iteration (ExIt)

We now define a family of exploration methods, which we refer to as Exploratory Iteration (ExIt) strategies. These methods directly exploit the recurrent structure of self-improvement tasks to actively generate and sample new task instances. Importantly, ExIt strategies seek to train the model for $K$-step self-improvement using only single-turn self-improvement tasks. This simplification is achieved by dynamically sampling starting solution iterates from a running buffer of the most informative solution iterates so far reached during training, allowing the single-step iteration tasks to accumulate to an effectively arbitrary depth during training. By decomposing extended self-improvement chains into one-step iteration tasks, this approach enables GRPO to credit-assign a dense reward to each indi-

---

**Algorithm 1:** Exploratory Iteration (ExIt)

Init policy $\pi_\theta$ and an empty task buffer $\mathcal{B}$ of size $N$
**while** *training* **do**
    **if** $|\mathcal{B}| \geq B_{\min}$ *and* `random()` $< p$ **then**
        Sample instance $(m, k, \tau_t^k) \sim \mathcal{B}$
        Sample partial history $\tau_{t^-}^{k'}$ for turn $t^- \leq t$
        Use instance $m' \leftarrow (m, k'+1, \tau_{t^-}^{k'})$
    **else**
        Sample base task $m' \sim \mathcal{M}$
    **end**
    Self-iterate from $m'$ to produce $m^+$
    Evaluate score $S$ for group rollouts of $m^+$ and update $\pi_\theta$ using these rollouts
    Update $\mathcal{B}$ with $m^+$ and $m'$ with score $S$
**end**

---

vidual self-improvement step, while naturally presenting each self-improvement task instance in an autocurriculum of improvement steps of progressively greater depth.

Concretely, new task instances are generated via two kinds of transformations, each acting on a task history $\tau_t^k$ up to $k$ self-iteration steps on the response for turn $t$ for task instance $m$. Here, $\tau_t^0$ corresponds to the history ending with the initial response for turn $t$ (without self-improvement).

**Selection:** Sample a task $m'$ with turn history $\tau_t$ from the self-improvement task buffer $\mathcal{B}$. Then, sample a random turn prefix containing the first $t^- \leq t$ turns from $\tau_t$, ending with the $k'$-th iterate of the response for turn $t^-$. Use this partial history $\tau_t^{k'}$ to create a new self-improvement task $m^- = (m, k'+1, \tau_{t^-}^{k'})$.

**Expansion:** Given a turn history $\tau_t^k$, take a self-iteration step from the last-turn response, resulting in an extended, complete history $\tau_{T_m}^{k+1}$. Expansion corresponds to self-improvement task $m^+ = (m, k+1, \tau_{T_m}^k)$.

Let $\Delta = G(\tau_{T_m}^{k+1}) - G(\tau_{T_m}^k)$ be the difference in quality measure between successive iterates. Under ExIt, the reward for self-iteration steps is $\max\left(0, \Delta/\Delta_{\max}\right)$, where $\Delta_{\max}$ is the largest possible improvement from

the previous iterate. The reward for base tasks remains the same. While GRPO already normalizes reward scales, we apply this explicit normalization to ensure variance-based learnability scores remain comparable.

At the start of each training episode, the task instance is first sampled via selection over the task buffer with probability $p$ if the task buffer is above a minimum size, and otherwise uniformly at random from the base task space $\mathcal{M}$. The one-step rollout from the sampled task instance then naturally performs expansion, and we subsequently update the task buffer with the new self-improvement instance resulting from expansion.

In practice, when updating the task buffer with a new instance with history $\tau_t^k$, we precompute and insert all instances corresponding to the per-turn partial histories of $\tau_t^k$ that can be sampled via selection, when performing this buffer update. To limit the size of the task buffer, we take the simple approach of assigning a priority score to each task instance and keeping only the top $N$ such partial histories with the highest score. In this work, we use a learnability score, discussed in Section 4.1. We describe extensions of the expansion step to promote exploration in Section 4.2. The general ExIt approach is depicted in Figure 2, and the pseudocode for our approach, simplified for the case of a single rollout per training iteration, is provided in Algorithm 1. Our self-iteration prompt templates are provided in Appendix B.

## 4.1 Selecting for learning potential

To enforce a finite capacity for the task buffer and focus training on only the most useful task instances, we keep only the top $N$ task instances based on learning potential, measured by the variance of the final quality measure (e.g. total return) of each rollout in a GRPO group: $S = \mathbf{var}(\mathbf{r})$, where the quality measure is rescaled to ensure all values are in the range $[0, 1]$. The variance metric extends the *learnability* measure studied as a prioritized sampling metric for binary success outcomes (Vinyals et al., 2019; Rutherford et al., 2024) to general scalar outcomes. In deterministic task instances (such as all tasks considered in this work), high variance in outcomes indicates the policy often succeeds and fails, and thus the potential for improvement (i.e. learning). We leave modifications of variance-based learnability metrics for stochastic environments to future work. As this learnability metric is simply a function of GRPO's group rollouts, using it for prioritized sampling of training instances results in a *compute-equivalent* autocurriculum method.

When updating the buffer with a self-iteration task $m^+$ based on expanding a previous instance $m'$, we do not have a learnability score for $m^+$. In practice, we initialize $m^+$ with the same score as the empirical score for $m'$, with the assumption that higher-variance starting points will tend to lead to subsequent high-variance points and vice versa. For example, if the policy can always succeed on a self-iteration task $m'$, its learnability will be 0, which is the expected score assignment for $m^+$, which starts from the correct solution.

If the task buffer is at capacity (i.e. $|\mathcal{B}| = N$), a task instance is only inserted into the buffer if its score is greater than or equal to the instance with the lowest score in the buffer, which is replaced by the inserted instance. When sampling training instances from the buffer, the $i$-th instance in the buffer with score $S_i$, for $i \leq N$, is assigned a probability mass equal to $\exp(S_i \kappa) / \sum_{j=1}^{|\mathcal{B}|} \exp(S_j \kappa)$, where $\kappa$ is the inverse temperature.

## 4.2 Expanding for divergence

As ExIt accumulates new self-improvement tasks based on the model's previous responses, the resulting task diversity strongly depends on the diversity of the model's outputs. To counteract RL's tendency to reduce output diversity, we incorporate directly diversity-seeking components:

**Divergent improvements.** With probability $p_{\mathrm{div}}$, a self-iteration step becomes a *self-divergence* step, whereby the policy is prompted to improve on a previous solution while diverging significantly from it (See Appendix B for the self-divergence prompt). We find divergence steps to induce meaningfully different responses from the model, resulting in increased task-space coverage when integrated into an ExIt strategy.

**Multiplicative diversity bonus.** Given an embedding space and a GRPO rollout group, compute a diversity score $d_i$ for the solution produced by the $i$-th rollout as the solution's embedding distance to the group centroid embedding $\bar{e}$, normalized by the range of distances within the group:

$$d_i = \frac{\|e_i - \overline{e}\|}{\max_j(\|e_j - \overline{e}\|) - \min_j(\|e_j - \overline{e}\|)}. \tag{6}$$

Following the approach in Chung et al. (2025), these diversity scores serve as coefficients that multiply each group-relative advantage as defined in Equation 3, thereby relatively upweighting more successful trajectories and downweighting poor-performing trajectories by how much they diverge relative to the group.

## 5 Experiment setting and results

We investigate the effectiveness of ExIt at inducing inference-time self-improvement behavior in several task domains, which together test the benefits of ExIt in improving test-time self-improvement in a variety of inference settings. These domains include mathematical reasoning (a standard, single-turn setting), tool-use (a multi-turn setting), and machine learning engineering tasks based on real Kaggle competitions (a setting where the LLM is typically run within a search scaffold).

### 5.1 Task domains

**Competition math.** We train on a randomly-sampled 1280-problem subset of an open dataset of competition math problems (Luo et al., 2025; Li et al., 2024) and evaluate models on several held-out test sets via math-verify (Kydlíček, 2025): MATH500 (Hendrycks et al., 2021), AMC12 2022/2023, AIME 2024/2025, the Minerva test split (Lewkowycz et al., 2022), and the text-only samples from the OlympiadBench test split (He et al., 2024). We specifically make use of this smaller training subset (epoched after every 10 GRPO iterations) in order to test the impact of ExIt in augmenting the training set with additional task instances.

**Multi-turn function calling.** The multi-turn-base split of the Berkeley Function Calling Leaderboard (BFCLv3) benchmark (Patil et al., 2025) provides a set of 200 tasks simulating multi-turn user interactions that require the LLM assistant to make a series of function calls based on API modules spanning domains including travel booking, stock price analysis, and file system management. We split these tasks into equal-sized train and test splits of 100 task instances, using stratified sampling to ensure each split shares a similar fraction of instances of different total turn counts and API sets. Following the modified environment logic in Zhuang et al. (2025), a trajectory is considered correct if both 1) its sequence of function calls leads to the final ground-truth state, and 2) the ground-truth sequence of function calls is a subset of the sequence. During RL fine-tuning, this correctness check serves as the verified reward function. This task allows us to evaluate ExIt's impact on a natively multi-turn task.

**ML engineering.** Coding naturally benefits from inference-time self-improvement, where it takes the form of debugging and optimization of previously output code solutions. We evaluate ExIt's impact on improving coding performance in MLE-bench (Chan et al., 2024), based on historical Kaggle competitions, by adapting this benchmark into an RL environment. The state-of-the-art solutions in MLE-bench rely on a search scaffold—a program that repeatedly calls the LLM in order to arrive at a final solution, by iterating on intermediate outputs (Jiang et al., 2025; Liu et al., 2025a; Toledo et al., 2025; Zhao et al., 2025b). We make use of this more challenging domain as a testbed for the ability of ExIt strategies to improve model performance within a search scaffold for code iteration (with error traces as execution feedback). We train on three tasks: detecting-insults-in-social-commentary, spooky-author-identification, and random-acts-of-pizza. At test time, we run the model within a greedy-search scaffold on held-out tasks: jigsaw-toxic-comment-classification-challenge, nomad2018-predict-transparent-conductors, and aerial-cactus-identification.

We provide additional experiment details and our choice of hyperparameters in Appendix A.

### 5.2 Baselines and ablations

In each task domain, we compare models trained via ExIt and its ablations against the original model before and after standard GRPO fine-tuning, via a heavily-modified fork of open-instruct (Wang et al., 2023). Specifically, we look at ablated variants of the "full" ExIt strategy: using only the learnability curriculum (i.e. only the selection transformation), additionally using self-improvement steps (Improve),

| Method | Math − All (%) | $\Delta_{16}$ | Tool-use (%) | $\Delta_4$ | MLE-bench (%) | $\Delta_{16}$ |
|---|---|---|---|---|---|---|
| Base model | 17.4±0.7 | -0.4 | 60.5±0.4 | +0.1 | 4.2±1.0 | +2.4 |
| GRPO | 18.7±0.4 | +1.1 | 73.5±1.5 | -0.5 | 48.0±0.7 | +9.1 |
| + curriculum | 18.8±0.7 | +0.9 | 75.4±1.2 | +1.0 | 53.0±9.7 | +11.9 |
| + self-improve step (Improve) | 19.6±0.7 | +1.2 | 75.5±1.2 | +3.4 | 47.8±2.5 | +9.4 |
| + divergent step (Diverge) | **20.1±0.7** | +1.6 | 76.8±2.2 | +0.6 | **57.3±7.3** | +10.1 |
| + diversity bonus (Full ExIt) | **20.4±0.5** | +2.0 | **76.4±1.0** | +1.2 | **58.6±2.5** | +8.4 |

Table 1: Evaluation of ExIt against its ablations and the GRPO baseline on held-out task instances. Math results are averaged over all test splits. All results are the mean and std of performance after 16 improvement steps over 3 training runs, alongside the net percentage of improvement ($\Delta_K$) between initial response and final response after $K$ self-improvement steps.

additionally using self-divergence steps (Diverge), and the full method additionally using the embedding-based diversity bonus (full ExIt). For the math domain, we use Llama-3.2-3B-Instruct (Meta AI, 2024), as its baseline performance leaves considerable room for improvement. We compute embeddings for the diversity bonus using a BERT model fine-tuned on an academic math dataset (Devlin et al., 2019; Steinfeldt & Mihaljević, 2024). For the more challenging tool-use and ML engineering domains, we use the best model at the 7B parameter scale from the Qwen2.5 series (Team, 2024) for each task. For tool-use, we fine-tune Qwen2.5-7B-Instruct, and for ML engineering, DeepSeek-R1-Distill-Qwen-7B (Guo et al., 2025). We embed both tool-use responses and code solutions using the 400M parameter CodeXEmbed model (Liu et al., 2024).

### 5.3 Self-improvement at inference time

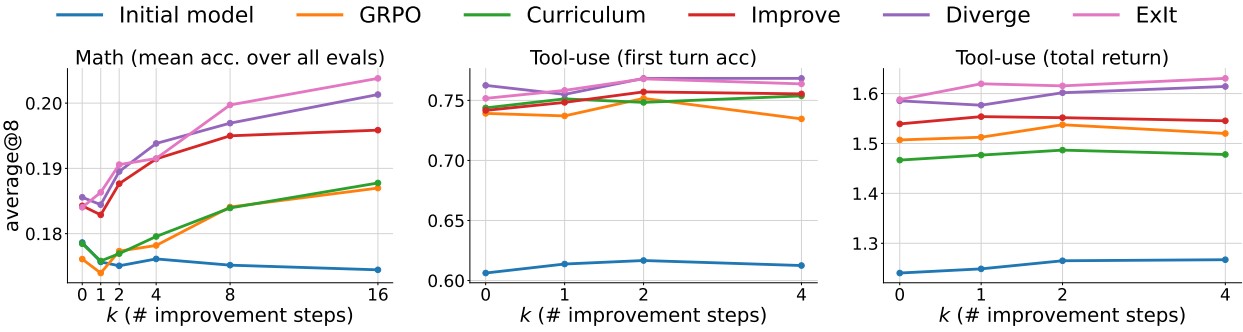

Figure 3: Left: Mean accuracy on all math test splits. Center: Mean first-turn accuracy on the multi-turn tool-use test split. Right: Mean total task return on the multi-turn tool-use test split. Results are avg@8 values across 3 training runs per method.

We measure the ability of a model to iteratively self-improve its responses at inference-time, with a budget of $K$ self-improvement steps. For each turn $t$, at self-improvement step $0 < k \leq K$, the model is prompted to provide an improved response given the original request (i.e. from the user) and the model's latest response iterates from all previous turns up to and including the current turn. The self-improvement results in Table 1 and Figures 3 – 5 show that across task domains, ExIt strategies directly using exploration mechanisms (Diverge's prompt-based exploration and the full ExIt strategy's use of an exploration bonus) consistently improve the model's ability to self-improve its solutions, with respect to the original model and after fine-tuning with GRPO. Notably, ExIt strategies also result in higher initial solution quality, while the curriculum-only baseline tends to achieve comparable performance with standard GRPO, indicating that augmented task-set diversity via selection and expansion transforms can result in more robust policies. We provide example self-iteration steps encountered during training in Appendix D.

In Figure 4, we see that across the held-out math splits, Improve, Diverge, and the full ExIt method all lead to a higher net number of successfully corrected solutions over 16 self-improvement steps, accumulating over 170 successful corrections across all held-out problems. On the majority of test splits (all except AIME

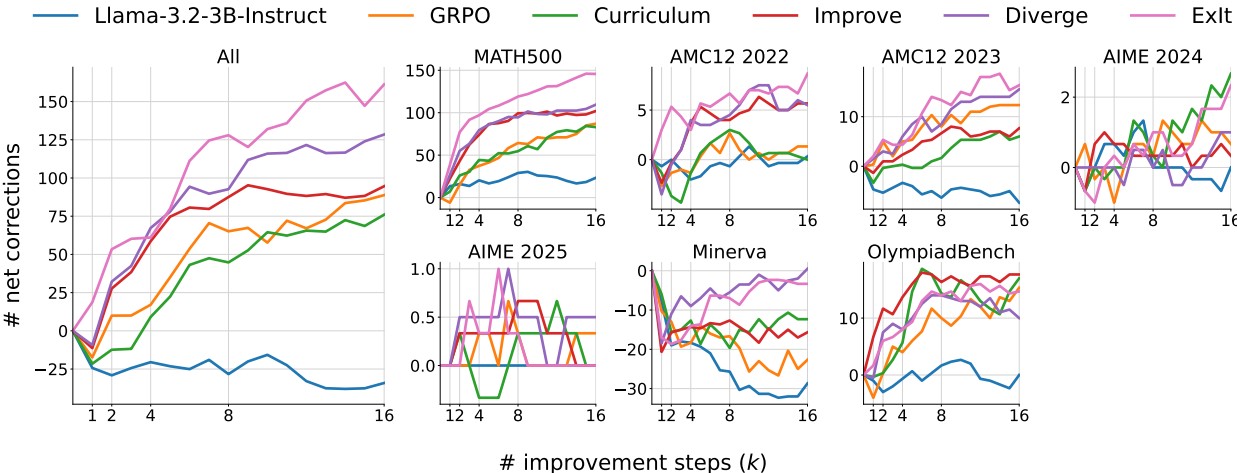

Figure 4: Net corrections across held-out math splits, computed over 8 samples per problem per checkpoint, averaged over 3 training runs per method (and an equivalent # samples/problem for Llama-3.2-3B-Instruct).

2025, where all methods fail to meaningfully self-improve), ExIT and DIVERGE continue to net additional corrections even after 16 self-improvement steps, despite the mean self-iteration depth remaining under two steps throughout training (see Figure 6).

In the multi-turn tool-use domain, we find ExIT strategy policies lead to greater performance, both before and after self-improvement steps compared to GRPO and the initial model, which both achieve worse performance after self-iterating over $K = 4$ steps. Notably, we see that performance ranking on first-turn accuracy does not always correspond to the ranking on total return—the latter a function of all-turn accuracy—with the curriculum-only baseline performing worse than the GRPO baseline in all-turn return, while performing better on first-turn accuracy. We see that the additional exploration mechanisms employed by DIVERGE and the full ExIT method enable the curriculum-based approach to exceed the performance of GRPO.

In MLE-bench, we find that both DIVERGE and the full ExIT method lead to the highest normalized performance across train and test tasks, while IMPROVE and the curriculum-only baseline attain similar performance as the GRPO baseline. This result highlights how even standard GRPO fine-tuning on a small set of MLE-bench tasks can transfer to held-out tasks. Our approach then further improves the quality of solutions discovered by the fine-tuned model within the multi-step greedy-search scaffold. This result shows that ExIT can be an efficient approach for fine-tuning LLMs for use in extended multi-step scaffolds at test-time, by producing and learning from purely single-step iterations at training time.

Appendix C further reports how success rate evolves per task split for the math and MLE-bench domains.

## 5.4 Emergent complexity during training

By selectively sampling task instances with the highest group return variance, ExIT strategies induce an adaptive curriculum of task instances that prioritizes those that are most learnable for the model at any time, without need for an externally-specified task ordering. Figures 5 and 6 show the curriculum-driven evolution of various task complexity metrics that intuitively align with task difficulty. Across domains, we find ExIT strategies tend to lead to greater complexity metrics than the curriculum-only baseline, with the direct exploration variants (DIVERGE and full ExIT) most often achieving the highest values. In the math domain, we see that the mean number of ground-truth solution steps of sampled instances increases over time, indicating the emergent curriculum naturally drives towards more challenging problems over the course of training. Similarly, in the multi-turn tool-use domain, the mean number of ground-truth tool-call steps of sampled task instances increases over time. Here we also see that the mean starting turn of a replay trajectory progressively increases during training, indicating that as the model becomes more adept at handling tool calling at earlier steps, the frontier of learnability moves deeper into the solution sequence. While ExIT variants also increase sample depth—the number of self-iteration steps already taken

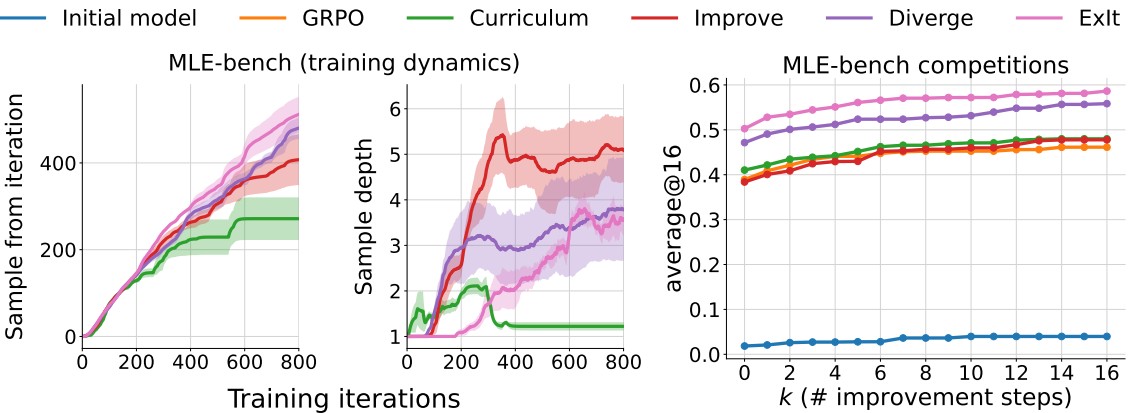

Figure 5: Left: Emergent curriculum over the sampled history's recency and starting depth during MLE-bench training. Right: Normalized MLE-bench scores achieved by each method over all train and test tasks via increasing greedy-search budgets (mean over 3 training runs).

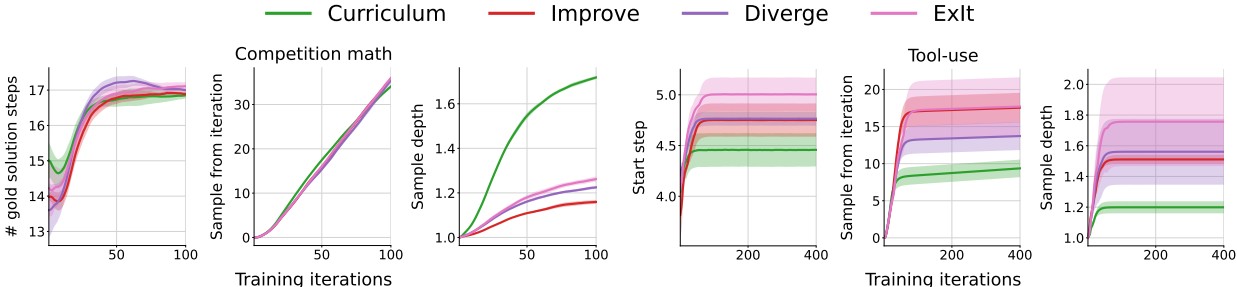

Figure 6: The evolution of various task instance properties during training in the math domain (left) and the multi-turn tool-use domain (right). Results are based on mean and std over 3 training runs.

on the starting response—more than the curriculum-only baseline, we find that the rank ordering among methods is inconsistent. This may be due to how self-iteration does not guarantee subsequent iterates deviate substantially from previous iterates in terms of difficulty for further improvement.

## 5.5 Exploratory iteration increases task diversity

We now analyze the diversity of task instances sampled by each method during training. Figure 7 shows the number of distinct training task instances relative to the base training set used by GRPO. In the tool-use domain, we count the number of unique task prompts, corresponding to the contents of the last user message in the message history at the start of each episode. For MLE-bench, we count the number of unique starting code solutions. We find that the curriculum-only baseline heavily shrinks the number of distinct training instances encountered, indicating that prioritized sampling leads to a high repetition of training instances. This reduction of task diversity may explain

| Method | Cosine dist. ($\uparrow$) | L2 dist. ($\uparrow$) |
|---|---|---|
| Uniform | N/A | N/A |
| Curriculum | N/A | N/A |
| Improve | 0.10 | 0.05 |
| Diverge | 0.11 | 0.06 |
| ExIt | **0.13** | **0.07** |

Table 2: Divergence metrics for training instances encountered for each method in MLE-bench tasks.

the relative underperformance of this baseline compared to ExIt variants, whose self-iteration steps have the effect of recovering a substantial amount of this lost diversity. In the case of the full ExIt, we see this improved diversity under a curriculum corresponds to improved performance on held-out instances. In the base training distribution used by GRPO, MLE-bench tasks all begin with the same empty Python script template, so we see a drastic increase in number of distinct starting code solutions under ExIt strategies. The UMAP (McInnes et al., 2018) in Figure 7 further highlights the differences between the task sets induced

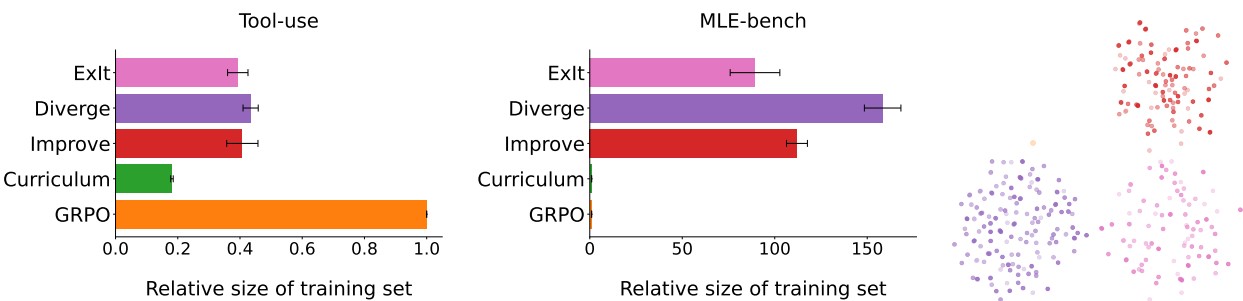

Figure 7: Relative number of distinct training task instances under each method for the multi-turn tool-use domain (left) and MLE-bench (middle). The UMAP projection of the distinct starting programs observed during MLE-bench training shows ExIt strategies increase the diversity of the training distribution (right).

by ExIt variants and the base task set—a single point in the CodeXEmbed embedding space. As reported in Table 2, the directly novelty-seeking ExIt variants lead to greater mean pairwise cosine and L2 distances among discovered task instances in the same embedding space, with the full ExIt attaining the greatest mean pairwise distances.

## 6 Related works

**Inference-time self-improvement.** Prior works consider variants of the inference-time self-improvement setting. Shinn et al. (2023) introduces a prompting-based approach for adapting based on previous trials of the same task with explicit feedback. Kumar et al. (2024) study directly training for a single step of self-improvement with optional feedback, which is a special case of the more general $k$-step setting considered in this work. They train a model to self-correct an initial previous solution in two training stages, with the first focusing on generating diverse initial solutions, and the second on self-correction. In contrast, ExIt does not require staged training and trains only on single-step iterations rather than both iterations together, while also generalizing beyond two self-improvement steps at test-time. Bauer et al. (2023), directly train an RL agent to reflect over multiple trials in a procedurally-generated game. Importantly, ExIt performs self-iteration per timestep (i.e. turn), a more granular contextual unit of iteration than the full episodic trial considered in these previous works. Per-timestep self-iteration is often a more suitable granularity for LLM tasks, in which each timestep typically maps to a full turn of a meaningful interaction. ExIt strategies can thus be complementary to the trial-based self-improvement setting. Another complementary research direction focuses on directly training improved verifiers for providing feedback for self-improvement (Gou et al., 2023; Dou et al., 2024; Yuan & Xie, 2025).

**Task-space exploration.** ExIt can be viewed as a form of Prioritized Level Replay Jiang et al. (2021b;a, PLR) that performs task-exploration at the turn level, by treating self-iteration from the response at each previous turn as a task instance in itself. Our task expansions can be seen as a kind of "mutation" operator on the task buffer, likening our approach to a version of PLR that explores the task space via evolutionary search (Parker-Holder et al., 2022). By adapting these ideas to self-improvement decision processes, ExIt demonstrates how task and trajectory-level exploration naturally coincide in this setting. Outside of the self-improvement setting, previous methods consider reset-based exploration at a trajectory level (Ecoffet et al., 2019; Kazemnejad et al., 2024; Foster & Foerster, 2025; Zheng et al., 2025). Unlike these approaches, ExIt modifies the specific task from the reset state (e.g. transforming it into a self-improvement or self-divergence task), while remaining compute-equivalent. In contrast to prior methods for dynamic training-task generation (Sukhbaatar et al., 2017; Wang et al., 2019; 2020; Dennis et al., 2020; Zhang et al., 2023; Faldor et al., 2024; Zhao et al., 2025a), ExIt forgoes a generator module and pursues open-ended task-space exploration (Jiang et al., 2023) by upcycling the learning agent's own trajectories into new task instances. The novelty bonus used by the full ExIt strategy adapts the approach in Chung et al. (2025) to GRPO. Our results show such bonuses can improve not just output diversity, but also task performance.

## 7 Limitations and future work

Our experiments feature a maximum training turn-history length of ten (in the tool-use domain). In domains like math and MLE-bench, where the responses tend to feature longer outputs, a multi-turn history can quickly exhaust the available context length. Extending EXIT to longer multi-turn training contexts serves as important future work, e.g. by compacting the context or using tool-based retrieval over the turn history. Though training FLOPS are still concentrated on base LLM computations, the use of an embedding model to compute response diversity adds extra overhead, making full EXIT only approximately compute-equivalent—though with minimal impact on training time, as this extra computation is parallelized with the backward pass. Evaluating full EXIT under cheaper metrics, e.g. based on edit-distance, would be valuable follow-up work. Moreover, our experiments use EXIT to drive parameter-space optimization, when the same strategies can also be applied to optimize the system prompt or task prompt template to induce greater self-improvement capabilities. Lastly, our experiments each focus on a single task domain and make use of an older generation of open models. The improved base reasoning capabilities of more recent models may enable them to benefit more from EXIT -style fine-tuning and exhibit greater transfer to unseen tasks. We leave application of EXIT to the latest base models for future work.

## 8 Conclusion

We introduced EXIT, an approach for RL fine-tuning an LLM for multi-step inference-time self-improvement via purely single-step self-improvement tasks. The core mechanisms behind EXIT adapt ideas from autocurriculum methods and trajectory-level exploration for RL to the $K$-step self-improvement problem setting. Importantly, as seen in our MLE-bench experiments, this problem setting maps closely to the common paradigm of performing LLM inference within a search scaffold. Our results show that EXIT indeed can result in improved downstream performance when the fine-tuned model is used to drive a search scaffold. Improvements to the self-iteration prompts (i.e. improve and diverge) as well as extensions of this set of operations to additional search-specific actions may further improve the impact of our approach in downstream scaffold-based applications. We believe self-supervised turn-level exploration, as embodied by the EXIT design, is a promising direction for greatly enhancing the efficacy of RL fine-tuning for LLMs.

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

# A  Additional experiment details

## A.1  Reinforcement learning system

Our GRPO trainer is based on a heavily-modified fork of open-instruct, which makes use of asynchronous GRPO updates, with separate training and inference ranks. In our modified trainer, training and evaluation rollouts are conducted via task-specific multi-turn RL environments. LLM inference is performed via asynchronous requests to a vLLM instance (AsyncLLM) of the model, whose weights are updated to the latest training weights after every GRPO training iteration.

## A.2  Competition math

Our RL training system represents each math task instance as a single-turn RL environment instance. The reward function for each instance is based on the outcome of comparing the extracted LLM solution with the ground-truth answer using math-verify (Kydlíček, 2025). The reward is 1 for a correct answer, and 0 otherwise. We use the same verification-based reward for self-improvement and self-divergence steps, effectively rewarding self-iteration responses within a GRPO group for improving an incorrect solution (or maintaining a correct solution) and penalizing responses that lead to an incorrect solution.

## A.3  Multi-turn tool-use

We follow the set up in Zhuang et al. (2025), adapting the BFCLv3 multi-turn-base task format into a multi-turn RL environment. Each task requires the LLM to fulfill a sequence of possibly multiple user requests by executing a sequence of tool calls, with the valid function signatures provided as JSON schemas in the system prompt. At each turn the LLM policy can either output a tool-call in JSON format, surrounded by `<tool>` tags, `<TASK_FINISHED>` to indicate the current user request has been fulfilled, or `<TASK_ERROR>` to indicate that there is unrecoverable error. We configure vLLM to stop inference after generating any of these kinds of responses. The LLM policy receives a reward of 1.0 for correctly outputting a sequence of tool calls that both includes the ground-truth sequence and results in the underlying program state equivalent to the expected program state after executing the ground-truth solution sequence. Like in the math domain, we also use this verification-based reward for episodes containing self-iteration steps.

Episodes in this environment are not guaranteed to share the same length, leading to the possibility of some trajectories being truncated when packing input-completion pairs into a training batch. To avoid this truncation, we trained on the full rendered message history with non-assistant messages masked out. To further stabilize training against overly long responses, we perform overlong filtering (Yu et al., 2025), setting the gradient to zero for all responses lacking any of the valid response types described above. We further found using the normalized objective from Liu et al. (2025b) to improve learning stability.

To limit training trajectory lengths, we use only the first user request and provide the agent a budget of 10 turns. During testing, we provide the agent with up to 60 turns to fulfill all user requests.

## A.4  MLE-bench

| Competition | Score type | Worst score | Best score |
|---|---|---|---|
| random-acts-of-pizza | AUC | 0.0 | 1.0 |
| spooky-author-identification | Cross-entropy loss | 1.5 | 1.0 |
| detecting-insults-in-social-commentary | AUC | 0.0 | 1.0 |
| nomad2018-predict-transparent-conductors | RMS error | 1.0 | 0.0 |
| jigsaw-toxic-comment-classification-challenge | AUC | 0.0 | 1.0 |
| aerial-cactus-identification | AUC | 0.0 | 1.0 |

Table 3: Score ranges used for computing normalized scores (rewards) each MLE-bench competition.

We adapt the Kaggle-based ML engineering tasks from Chan et al. (2024) into a single-turn RL environment, parameterized by the competition ID. The task description in the prompt corresponds to the compact "obfuscated" markdown competition description. The user prompt for each task instance (i.e. competition) is shown in Appendix B, with a goal component varying based on if the starting code is buggy (in which case, the error trace is also provided below the code block). The prompt instructs the LLM to improve this code block, which defaults to a Python script with an empty main function, with comments indicating the environment variable storing the data directory path (These paths are also described in the prompt and provided to the environment during initialization). Under ExIt strategies, the starting code block is replaced with the code solution from the previous response sampled for self-iteration.

The reward function is computed by first extracting and executing the code solution from the LLM response in a sandbox environment, which produces a submission.csv file. We limit the maximum runtime to 5 minutes, as solutions for the competitions we use for this study can typically be trained in just a few minutes, and this limit prevents training from stalling due to the presence of inefficient scripts in a rollout batch. Then, `mlebench grade` is executed to provide a score $r$ based on a metric specific to each competition, e.g. AUC. To compute normalized scores that reside in $[0.0, 1.0]$, we define ranges corresponding to the worst and best score per competition and compute the reward as the normalized score as

$$r_{\text{norm}} = \frac{r - r_{\text{worst}}}{r_{\text{best}} - r_{\text{worst}}}.$$

As some competitions make use of error metrics like cross-entropy loss that are in principle unbounded, we clip the worst values to a reasonable finite value based on the historical submission results from human contestants for these competitions. See Table 3 for a listing of each competition score function and corresponding ranges used for reward normalization. This normalized reward is also used for self-divergence steps, while the self-improvement reward for the $k$-th iterate, $r_k$ is computed as the normalized difference between the new solution and previous solution iterate, where $r_k$ and $r_{k-1}$ are assumed to be normalized as described above:

$$r_{\text{imp},k} = \max\left(0, \frac{r_k - r_{k-1}}{1 - r_{k-1}}\right).$$

### A.5  Choice of hyperparameters

Training hyperparameters for experiments in Section 5 are shown in Tables $4-6$. All experiments set clipping coefficient $\varepsilon = 0.2$. Except for learning rate, we use the best GRPO hyperparameters when sweeping ExIt hyperparameters. We evaluate using the default sampling configs fo each model on Hugging Face Hub.

**Competition math.** We validate against a held-out set of 100 problems sampled from the MATH training dataset and sweep over learning rate in [5e-7, 1e-6, 5e-6], $\beta$ in [0.0, 0.001, 0.01], buffer size in [100, 256, 512, 1024], and self-iteration and divergence probabilities in [0.2, 0.5, 0.8].

| | GRPO | Curriculum | Improve | Diverge | ExIt |
|---|---|---|---|---|---|
| Learning rate | 1e-6 | | | | |
| Prompts per batch | 128 | | | | |
| Rollouts per prompt | 8 | | | | |
| Epochs per batch | 1 | | | | |
| KL coefficient, $\beta$ | 0.001 | | | | |
| Ref. update interval, $M$ | 100 | | | | |
| Ref. update $\alpha$ | 1.0 | | | | |
| Total buffer size | – | 512 | 512 | 512 | 512 |
| Min. buffer size | – | 128 | 128 | 128 | 128 |
| Self-iteration prob. | – | 0.5 | 0.5 | 0.5 | 0.5 |
| Divergence prob. | – | – | – | 0.2 | 0.2 |

Table 4: Hyperparameters for math experiments for each method and ablation.

**Multi-turn tool-use.** We select hyperparameters for GRPO based on a 25-task validation subset held-out from the training split. The final checkpoints train on the full 100 training tasks. For ExIt methods, we select hyperparameters based on those producing curricula plateauing with the highest starting step during training, treating this metric as a proxy for improved robustness over preceding steps. We sweep over learning rate in [1e-7, 5e-7, 1e-6, 5e-6], $\beta$ in [0.0, 0.001, 0.01], buffer size in [100, 320, 640], and self-iteration and divergence probabilities in [0.2, 0.5, 0.8]. Choices of # epochs, $M$, and $\alpha$ are based on Zhuang et al. (2025).

| | GRPO | Curriculum | Improve | Diverge | ExIt |
|---|---|---|---|---|---|
| Learning rate | 1e-6 | 5e-7 | 5e-7 | 5e-7 | 1e-6 |
| Prompts per batch | 16 | | | | |
| Rollouts per prompt | 8 | | | | |
| Epochs per batch | 2 | | | | |
| KL coefficient, $\beta$ | 0.001 | | | | |
| Ref. update interval, $M$ | 100 | | | | |
| Ref. update $\alpha$ | 1.0 | | | | |
| Total buffer size | – | 320 | 640 | 640 | 640 |
| Min. buffer size | – | 320 | 320 | 320 | 320 |
| Self-iteration prob. | – | 0.2 | 0.5 | 0.5 | 0.5 |
| Divergence prob. | – | – | - | 0.2 | 0.5 |

Table 5: Hyperparameters for multi-turn tool-use experiments for each method and ablation.

**MLE-bench.** We select hyperparameters that maximize improvement on training tasks over 16 greedy search steps and sweep over learning rate in [5e-7, 1e-6, 5e-6], $\beta$ in [0.0, 0.001, 0.01], GRPO $\alpha$ in [0.0, 0.5, 1.0], buffer size in [3, 24, 32, 64], minimum buffer size in [3, 8], and self-iteration and divergence probabilities in [0.2, 0.5, 0.8].

| | GRPO | Curriculum | Improve | Diverge | ExIt |
|---|---|---|---|---|---|
| Learning rate | 1e-6 | | | | |
| Prompts per batch | 4 | | | | |
| Rollouts per prompt | 8 | | | | |
| Epochs per batch | 1 | | | | |
| KL coefficient, $\beta$ | 0.001 | | | | |
| Ref. update interval, $M$ | 100 | | | | |
| Ref. update $\alpha$ | 0.5 | | | | |
| Total buffer size | – | 3 | 24 | 32 | 24 |
| Min. buffer size | – | 3 | 8 | 8 | 3 |
| Self-iteration prob | – | 0.5 | 0.5 | 0.5 | 0.5 |
| Divergence prob | – | – | – | 0.2 | 0.5 |

Table 6: Hyperparameters for MLE-bench experiments for each method and ablation.

## B  Prompts

We provide the prompt templates we use for self-iteration (self-improve and self-divergence) steps. The generic self-iteration prompts are shared across math and multi-turn tool use environments, while MLE-bench makes use of a separate prompt format, which contains additional task-specific information required by this domain. For the MLE-bench prompt template, the self-iteration task is specified by the "goal preamble" component of the prompt. Note that in this domain, self-improvement takes the form of a debug step when the previous solution iterate has errors (which are then included below the code block as a form of task feedback).

---

**Self-improve prompt**

```
Improve your current response to this request:

## Request:
{instruction}

## Current response:
{previous_response}
```

---

**Self-diverge prompt**

```
Consider your current response to this request, and provide another response that takes
an approach that SIGNIFICANTLY DIFFERS from the approach of the current response.

## Request:
{instruction}

## Current response:
{previous_response}

## Your new response format:
- First, provide a brief summary of the approach in the current response above.
- Next, state a meaningfully different approach that can be taken instead.
- Finally, provide your new response using this different approach.
```

---

**MLE-bench prompt**

```
You are a Kaggle grandmaster.
Your solution to this Kaggle challenge should achieve the highest medal possible, where the medal
rankings from best to worst are gold, silver, and bronze.
Your solution is not good enough if it does not medal.

Study the current version of solution.py:
```python
{initial_solution}
```{debug_info}

{goal_preamble}

# Task description

{task_description}

# Previews of files located at PUBLIC_RESOURCE_PATH:

{preview_contents}

Write a single file called solution.py that does the following:
```

---

```
    - Loads any relevant training data described in the task description.
    - Loads any relevant test data described in the task description.
    - It is important your code references the training and test data paths exactly as above,
      via these environment variables, which will already be set correctly for you.
    - Performs any arbitrary training and evaluation logic on this data split.
      For example, you might hold out some subset of this data for the purposes
      of k-fold cross validation for model and hyperparameter selection.
    - Finally, evaluates the final model on the test data and writes the results
      to a file in the current directory called submission.csv.

All files and directories described in the task description are located under the
PUBLIC_RESOURCE_PATH, whose value is always available via os.environ.get('PUBLIC_RESOURCE_PATH').
Besides solution.py, you may not create any extra files or modify any other files in the
directory.
Remember to define all necessary functions before the 'if __name__ == '__main__'' block.
Your code will be run on a machine with a single H100 GPU.

Respond with your full working python program for this task inside triple-tick delimiters ('''').
```

## MLE-bench improve goal preamble

Your goal is to improve this python script to better achieve the following task:

## MLE-bench debug goal preamble

Your goal is to debug this python script, so that it best achieves the following task:

## MLE-bench diverge goal preamble

Your goal is to propose a new python script that improves over this python script for the
following task.
Your improved script should take a radically different approach to solving the problem than the
one taken by the current python script.

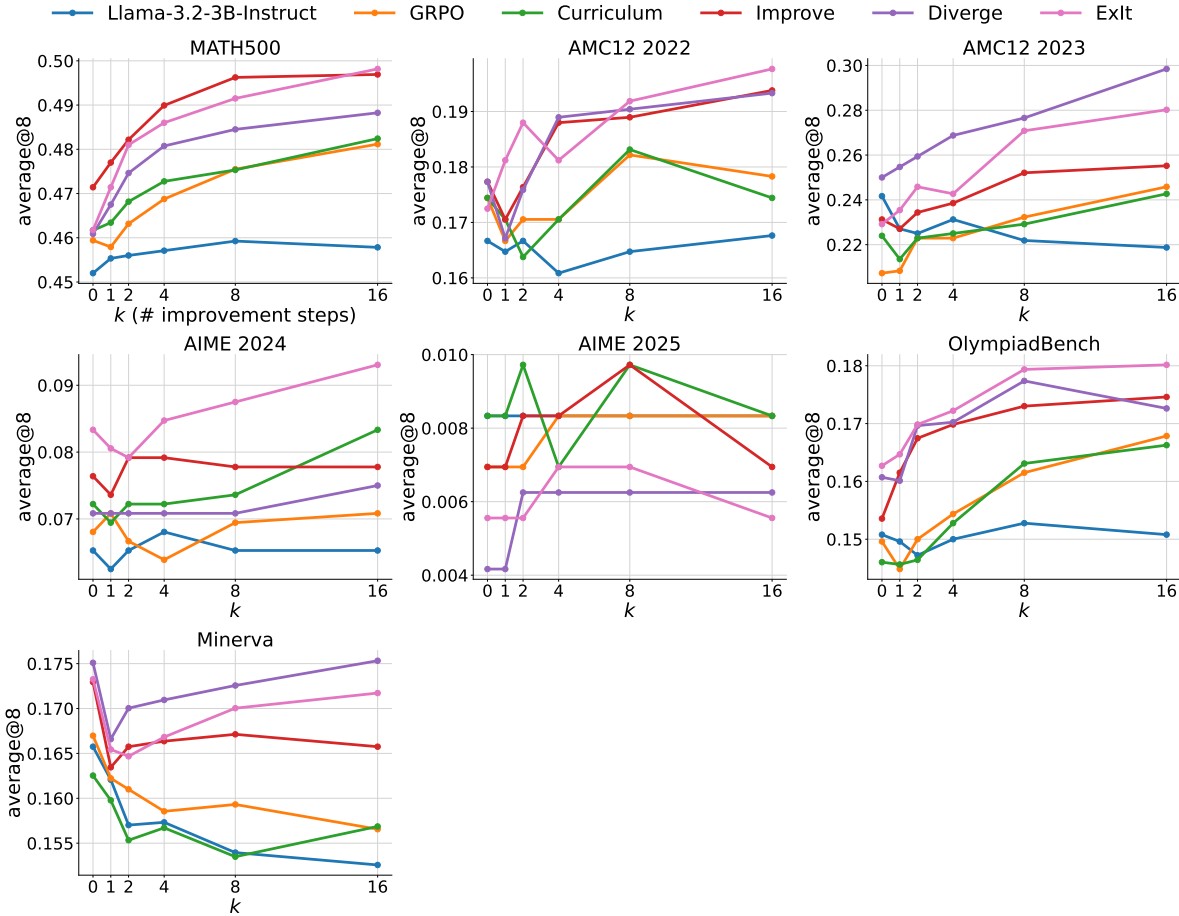

Figure 8: Net incorrect → correct improvements across all held-out math splits. Total corrections are averages computed over 8 samples per problem for each of 3 checkpoints per method, and an equivalent number of samples per problem for Llama-3.2-3B-Instruct.

## C  Additional experiment results

In Figure 8, we show the inference-time self-improvement performance over 16 steps for each test split in the competition math domain. Both DIVERGE and the full EXIT strategy produce improved performance across these test splits, with the exception of AIME 2025, where none of the methods perform well.

As we saw in Figure 7, EXIT strategies increase the effective number of distinct tasks encountered during training with respect to the curriculum-only baseline, which reduces task diversity by oversampling those task instances with higher return variance. In Table 7, we see that the base task set has slightly greater pairwise embedding distances relative to the augmented task set discovered by EXIT, under the embedding space of the 400M parameter CodeXEmbed model. We see EXIT strategies are able to maintain a comparable degree of variation to the base task set while increasing the number of tasks considered for prioritized sampling.

Figure 9 reports the mean greedy-search performance over 16 steps on the train and test competitions. Figure 10 shows greedy-search performance per competition. We see that for training competitions, the LLM policy rapidly converges to a specific solution, likely due to some degree of overfitting from directly training on the task instance, while test competitions see greater differences between initial and final solution performance after 16 greedy-search steps. The initial model is unable to effectively self-improve, and standard GRPO fine-tuning already induces some degree of self-improvement. EXIT strategies directly using step-level exploration further enhance the model's ability to self-improve.

| Method | Cosine dist. ($\uparrow$) | L2 dist. ($\uparrow$) |
|---|---|---|
| Uniform | 0.56 | 0.33 |
| Curriculum | 0.56 | 0.32 |
| Improve | 0.52 | 0.30 |
| Diverge | 0.48 | 0.28 |
| ExIt | 0.50 | 0.29 |

Table 7: Mean pairwise distances computed over the training task instances used by each method.

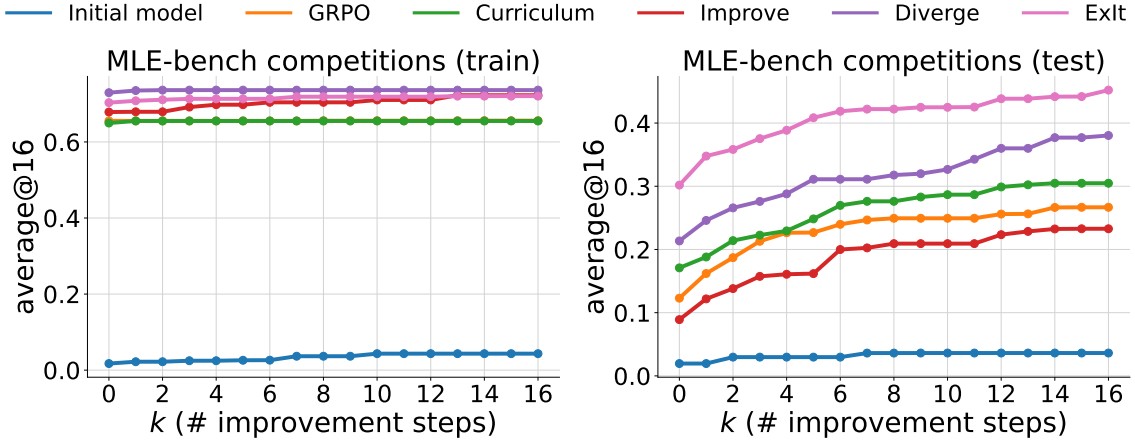

Figure 9: Performance of models trained via each method when run in a greedy-search scaffold on the MLE-bench train (left) and test (right) competitions.

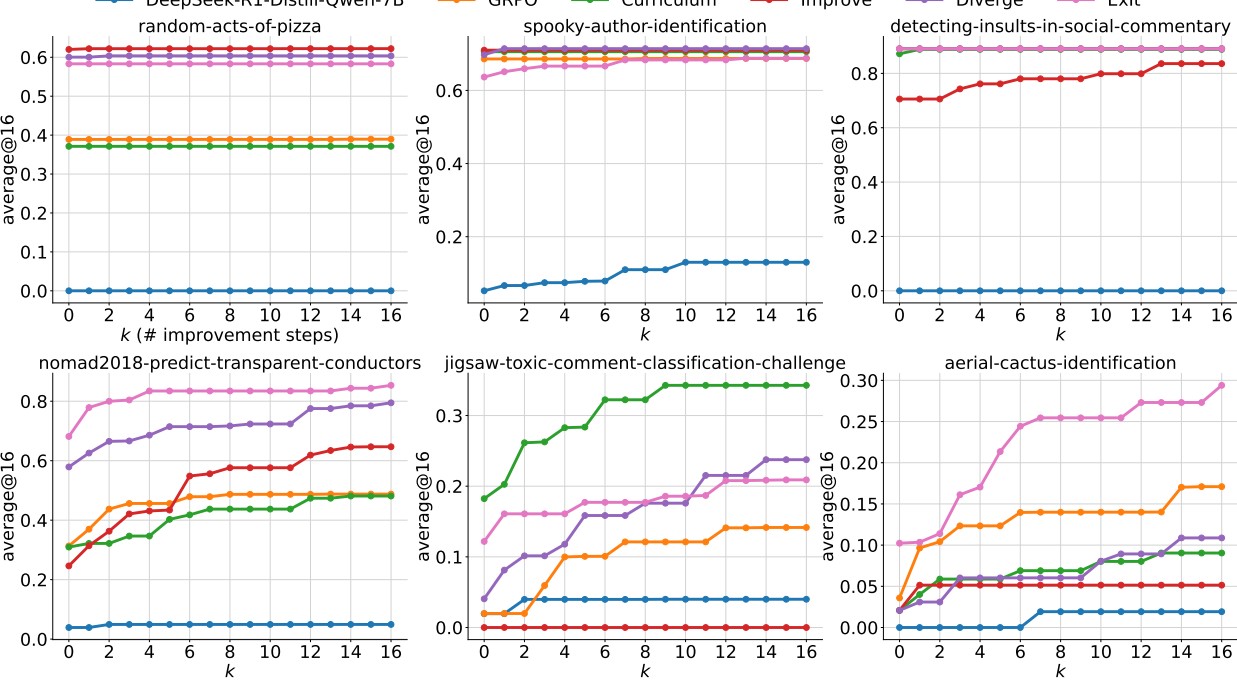

Figure 10: Improvement over 16 greedy-search steps using the initial model and the initial model fine-tuned using each method on train (top) and test (bottom) MLE-bench competitions.

# D    Example self-iteration steps

We provide examples of self-iteration prompts and responses encountered over the course of training via the full EXIT method. We see that the model can respond successively to both self-improvement and self-divergence prompts. Notably, we see in example from the tool-use domain that self-divergence can lead not only to new solution approaches, but also improvements to a previously faulty response.

---

**Improving on a math solution (self-correction)**

Improve your current response to this request:

## Request:
In a round-robin tournament with 6 teams, each team plays one game against each other team, and each game results in one team winning and one team losing. At the end of the tournament, the teams are ranked by the number of games won. What is the maximum number of teams that could be tied for the most wins at the end of the tournament?

## Current response:
## Step 1: Determine the total number of games played in the tournament.
In a round-robin tournament with 6 teams, each team plays one game against each of the other 5 teams. So, the total number of games played is the number of teams choose 2, calculated as C(6, 2) = 6! / (2!(6-2)!) = 15 games.

## Step 2: Determine the maximum number of wins a team can have.
Since each game results in one win and one loss, the total number of wins across all teams is equal to the total number of games played. Therefore, the maximum number of wins a team can have is the total number of games divided by the number of teams, which is 15 / 6 = 2.5. However, since a team cannot have a fraction of a win, the maximum number of wins a team can have is 2.

## Step 3: Determine the maximum number of teams that could be tied for the most wins.
For a maximum number of teams to be tied for the most wins, we need to maximize the number of teams with the maximum possible number of wins. Since each team can have at most 2 wins, the maximum number of teams that can be tied for the most wins is 5. This is because if all teams had the same number of wins, and a team had 2 wins, the remaining 5 teams could also have 2 wins if they each lost to the team with 2 wins, but this would mean the team with 2 wins would have played 5 games, which is not possible. However, we can achieve a tie with 5 teams if we consider that one team has 4 wins and the remaining 5 teams have 1 win each.

The final answer is: $\boxed{4}$

---

## Step 1: Determine the total number of games played in the tournament. In a round-robin tournament with 6 teams, each team plays one game against each of the other 5 teams. So, the total number of games played is the number of teams choose 2, calculated as C(6, 2) = 6! / (2!(6-2)!) = 15 games.

## Step 2: Determine the maximum number of wins a team can have. Since each game results in one win and one loss, the total number of wins across all teams is equal to the total number of games played. Therefore, the maximum number of wins a team can have is the total number of games divided by the number of teams, which is 15 / 6 = 2.5. However, since a team cannot have a fraction of a win, the maximum number of wins a team can have is 2.

## Step 3: Determine the maximum number of teams that could be tied for the most wins. For a maximum number of teams to be tied for the most wins, we need to maximize the number of teams with the maximum possible number of wins. Since each team can have at most 2 wins, the maximum number of teams that can be tied for the most wins is 5. This is because if all teams had the same number of wins, and a team had 2 wins, the remaining 5 teams could also have 2 wins if they each lost to the team with 2 wins, but this would mean the team with 2 wins would have played 5 games, which is not possible. However, we can achieve a tie with 5 teams if we consider that one team has 4 wins and the remaining 5 teams have 1 win each.

The final answer is: $\boxed{5}$

---

**Diverging from a math solution**

Consider your current response to this request, and provide another response that takes an approach that SIGNIFICANTLY DIFFERS from the approach of the current response.

## Request:
A 50-card deck consists of 4 cards labeled " $i$ " for $i = 1, 2, \ldots, 12$ and 2 cards labeled " 13 ". If Bob randomly chooses 2

cards from the deck without replacement, what is the probability that his 2 cards have the same label?

## Current response:
## Step 1: Calculate the total number of ways to choose 2 cards from a 50-card deck without replacement. The total number of ways to choose 2 cards from 50 cards without replacement can be calculated using combinations. This can be expressed as $\binom{50}{2} = \frac{50!}{2!(50-2)!} = \frac{50*49}{2} = 1225$.

## Step 2: Calculate the number of ways to choose 2 cards with the same label. There are 12 cards labeled "i" and 2 cards labeled "13". We need to calculate the number of ways to choose 2 cards with the same label for each type of card.

## Step 3: Calculate the number of ways to choose 2 cards with the same label for the 12 labeled "i" cards. For each of the 12 labeled "i" cards, we can choose 2 cards out of the 4 cards of the same label. This can be calculated using combinations as $\binom{4}{2} = \frac{4!}{2!(4-2)!} = \frac{4*3}{2} = 6$. Since there are 12 different labels, the total number of ways to choose 2 cards with the same label for the "i" cards is $12 * 6 = 72$.

## Step 4: Calculate the number of ways to choose 2 cards with the same label for the 2 labeled "13" cards. For the 2 labeled "13" cards, we can choose 2 cards out of the 2 cards of the same label. This can be calculated using combinations as $\binom{2}{2} = \frac{2!}{2!(2-2)!} = 1$.

## Step 5: Calculate the total number of favorable outcomes. The total number of favorable outcomes is the sum of the number of ways to choose 2 cards with the same label for the "i" cards and the number of ways to choose 2 cards with the same label for the "13" cards. This is $72 + 1 = 73$.

## Step 6: Calculate the probability. The probability of choosing 2 cards with the same label can be calculated by dividing the total number of favorable outcomes by the total number of ways to choose 2 cards. This is $\frac{73}{1225}$.

## Step 7: Simplify the fraction. The fraction $\frac{73}{1225}$ can be simplified by dividing both the numerator and the denominator by their greatest common divisor, which is 1. This results in $\frac{73}{1225}$.

The final answer is: $\boxed{\dfrac{73}{1225}}$

## Your new response format:
- First, provide a brief summary of the approach in the current response above.
- Next, state a meaningfully different approach that can be taken instead.
- Finally, provide your new response using this different approach.

**Summary of the current approach:**
The current approach involves calculating the total number of ways to choose 2 cards from the deck and then calculating the number of ways to choose 2 cards with the same label. This is done by calculating combinations for each type of card (labeled "i" and "13") and summing up the results.

**A meaningfully different approach:**
A different approach can be taken by using the concept of complementary probability. This involves calculating the probability that the two cards do not have the same label and subtracting this from 1.

**New response:**

## Step 1: Calculate the total number of ways to choose 2 cards from the deck without replacement. The total number of ways to choose 2 cards from 50 cards without replacement can be calculated using combinations. This can be expressed as $\binom{50}{2} = \frac{50!}{2!(50-2)!} = \frac{50*49}{2} = 1225$.

## Step 2: Calculate the number of ways to choose 2 cards that do not have the same label. To calculate the number of ways to choose 2 cards that do not have the same label, we can calculate the total number of ways to choose 2 cards and subtract the number of ways to choose 2 cards with the same label.

## Step 3: Calculate the number of ways to choose 2 cards with the same label. This was calculated in the current response as $72 + 1 = 73$.

## Step 4: Calculate the number of ways to choose 2 cards that do not have the same label. The number of ways to choose 2 cards that do not have the same label is $1225 - 73 = 1152$.

## Step 5: Calculate the probability that the two cards do not have the same label. The probability that the two cards do not have the same label can be calculated by dividing the number of ways to choose 2 cards that do not have the same

label by the total number of ways to choose 2 cards. This is $\frac{1152}{1225}$.

## Step 6: Calculate the probability that the two cards have the same label. The probability that the two cards have the same label can be calculated by subtracting the probability that the two cards do not have the same label from 1. This is $1 - \frac{1152}{1225} = \frac{73}{1225}$.

## Step 7: Simplify the fraction. The fraction $\frac{73}{1225}$ can be simplified by dividing both the numerator and the denominator by their greatest common divisor, which is 1. This results in $\frac{73}{1225}$.

The final answer is: $\boxed{\frac{73}{1225}}$

---

**Improving a tool-use decision**

{ Message history in which the start engine call results in an error indicating that the doors must first be locked, with the assistant mistakenly asking the user to lock the doors first }

## Request:
I've completed the maintenance on my car and ensured the doors are unlocked. Everything, especially the tires, seems in good condition. Would you kindly assist in securing the remaining doors and initiate the engine in START mode? I want everything primed before I set off.

## Current response:
<TASK_ERROR>

<think> Based on the response, the doors need to be locked again. Let's go through the steps more carefully. </think>
<tool> "name": "lockDoors", "parameters": "unlock": false, "door": ["driver", "passenger", "rear_left", "rear_right"] </tool>

---

**Diverging from a tool-use decision**

{ Message history preceding this turn, in which the assistant incorrectly uses city names as arguments for the distance function }

## Request:
Could you provide me with a rough distance from Rivermist (zip 83214) to Greenway (zip 74532)? I'm eager to plan a visit and need to know the travel distance.

## Current response:
TASK_ERROR

## Your new response format:
- First, provide a brief summary of the approach in the current response above.
- Next, state a meaningfully different approach that can be taken instead.
- Finally, provide your new response using this different approach.

Summary of the previous approach: The previous approach attempted to use city names to get valid zip codes but was unsuccessful. Since direct zip code retrieval did not work, the previous attempt ended in error.

Meaningfully different approach: Instead of using city names, we can directly use the provided zip codes for distance estimation.

<think> We will use the provided zip codes (83214 for Rivermist and 74532 for Greenway) to estimate the distance between the two cities. </think>

<tool> "name": "estimate_distance", "parameters": "cityA": "83214", "cityB": "74532" </tool>

---

**Improving an MLE-bench solution**

Study the current version of solution.py:

```
import os
```

```
import pandas as pd
from sklearn.naive_bayes import MultinomialNB
from sklearn.feature_extraction.text import CountVectorizer

def main():
    # Load the training data
    train_data = pd.read_csv(os.path.join(os.environ['PUBLIC_RESOURCE_PATH'], 'train.csv
        '))
    test_data = pd.read_csv(os.path.join(os.environ['PUBLIC_RESOURCE_PATH'], 'test.csv')
        )

    # Vectorize the text data
    vectorizer = CountVectorizer()
    X_train = vectorizer.fit_transform(train_data['text'])
    X_test = vectorizer.transform(test_data['text'])

    # Train the model
    model = MultinomialNB()
    model.fit(X_train, train_data['author'])

    # Get probabilities for test data
    preds = model.predict_proba(X_test)

    # Save to CSV
    submission = pd.DataFrame({
        'id': test_data['id'],
        'EAP': preds[:,0],
        'HPL': preds[:,1],
        'MWS': preds[:,2]
    })
    submission.to_csv('submission.csv', index=False)

if __name__ == '__main__':
    main()
```

Your goal is to improve this python script to better achieve the following task:

{ task description }

{ thinking not shown to conserve space }

To improve the Kaggle solution, I'll modify the code to:

1. Use Laplace smoothing (alpha=1) to handle sparse features
2. Ensure proper probability normalization
3. Add logging for better debugging

Here's the improved code:

```python
import os
import pandas as pd
from sklearn.naive_bayes import MultinomialNB
from sklearn.feature_extraction.text import CountVectorizer
from sklearn.metrics import log_loss

def main():
    # Load training and test data
    train_data = pd.read_csv(os.path.join(os.environ['PUBLIC_RESOURCE_PATH'], 'train.csv
        '))
    test_data = pd.read_csv(os.path.join(os.environ['PUBLIC_RESOURCE_PATH'], 'test.csv')
        )

    # Vectorize the text data
    vectorizer = CountVectorizer()
    X_train = vectorizer.fit_transform(train_data['text'])
```

```
    X_test = vectorizer.transform(test_data['text'])

    # Train the model
    model = MultinomialNB(alpha=1.0)
    model.fit(X_train, train_data['author'])

    # Get probabilities for test data
    preds = model.predict_proba(X_test)

    # Save to CSV
    submission = pd.DataFrame({
        'id': test_data['id'],
        'EAP': preds[:,0],
        'HPL': preds[:,1],
        'MWS': preds[:,2]
    })
    submission.to_csv('submission.csv', index=False)

if __name__ == '__main__':
    import logging
    logging.basicConfig(level=logging.INFO)
    logger = logging.getLogger(__name__)

    logger.info("Starting␣the␣Kaggle␣solution")
    logger.info(f"Loading␣data␣from␣{os.environ['PUBLIC_RESOURCE_PATH']}")

    main()
```

This code:
1. Adds Laplace smoothing (alpha=1) to handle low-count features
2. Uses logging for better debugging
3. Ensures proper probability output
4. Handles the submission format correctly

## Diverging from an MLE-bench solution

Study the current version of solution.py:

```
import os
import pandas as pd
from sklearn.feature_extraction.text import TfidfVectorizer
from sklearn.naive_bayes import MultinomialNB

def main():
    # Load the training and testing data
    train_data_path = os.path.join(os.environ.get('PUBLIC_RESOURCE_PATH'), 'train.csv')
    test_data_path = os.path.join(os.environ.get('PUBLIC_RESOURCE_PATH'), 'test.csv')

    train_data = pd.read_csv(train_data_path)
    test_data = pd.read_csv(test_data_path)

    # Prepare the data
    X_train = train_data['Comment']
    y_train = train_data['Insult']

    # Vectorize the text data
    vectorizer = TfidfVectorizer()
    X_train_vec = vectorizer.fit_transform(X_train)

    # Train the model
    model = MultinomialNB()
    model.fit(X_train_vec, y_train)

    # Make predictions
    X_test = test_data['Comment']
    X_test_vec = vectorizer.transform(X_test)
```

```
    y_pred = model.predict(X_test_vec)

    # Save the results
    submission = pd.DataFrame({'Insult': y_pred})
    submission.to_csv('submission.csv', index=False)

if __name__ == '__main__':
    main()
```

Your goal is to propose a new python script that improves over this python script for the following task. Your improved script should take a radically different approach to solving the problem than the one taken by the current python script.

{ task description }

{ thinking not shown to conserve space }

To solve this problem, I'll create a simple machine learning model using logistic regression with TF-IDF vectorization. Here's the complete solution:

```python
import os
import pandas as pd
from sklearn.feature_extraction.text import TfidfVectorizer
from sklearn.linear_model import LogisticRegression
from sklearn.metrics import roc_auc_score

def main():
    # Load data
    train_path = os.path.join(os.environ.get('PUBLIC_RESOURCE_PATH'), 'train.csv')
    test_path = os.path.join(os.environ.get('PUBLIC_RESOURCE_PATH'), 'test.csv')

    train_data = pd.read_csv(train_path)
    test_data = pd.read_csv(test_path)

    # Vectorize comments
    vectorizer = TfidfVectorizer()
    X_train = vectorizer.fit_transform(train_data['Comment'])

    # Train model
    model = LogisticRegression()
    model.fit(X_train, train_data['Insult'])

    # Get predictions
    X_test = test_data['Comment']
    X_test_vec = vectorizer.transform(X_test)
    preds = model.predict_proba(X_test_vec)[:,1]

    # Save results
    submission = pd.DataFrame({
        'Insult': preds,
        'Date': test_data['Date'],
        'Comment': test_data['Comment']
    })
    submission.to_csv('submission.csv', index=False)

if __name__ == '__main__':
    main()
```

This code:
1. Loads the training and test data
2. Vectorizes the comments using TF-IDF
3. Trains a logistic regression model
4. Generates predictions and saves them to submission.csv

The model uses TF-IDF for feature extraction and logistic regression for prediction, which should provide reasonable AUC scores given the problem constraints.

