# OpenReview forum: "Bootstrapping Task Spaces for Self-Improvement"
_TMLR — Accepted by TMLR_

### Review · Reviewer_FpLj · 2025-09-30

**Summary Of Contributions:**

The paper introduces a way to train LLMs to exhibit multi-step self-improvement at inference-time, while avoiding the complexity to train on full reasoning chains. This is achieved by iteratively constructing a prioritized buffer of partial histories, ranked by a learnability score (variance associated with GRPO). Subsequently, one SELECTS from this buffer, and EXPANDS the buffer which each rollout.

This process is combined with self-diverging prompts and a multiplicative bonus to encourage diversity. Training is performed only on single-step self-iteration transitions, just at test time one generalizes to K-step self-improvement.

The strategy is named "Exploratory Iteration (EXIT)", and its efficiency is demonstrated over pure GRPO and other baselines on three tasks: competition math, multi-turn tool use, and ML engineering. This improvement is in parts explained by increases task-space diversity due to using the EXIT strategy.

**Additional Comments:**

What is a bit unlucky: There is already a method called "ExIt" (Expert Iteration, see
https://doi.org/10.48550/arXiv.1705.08439) in an related domain, so maybe changing the abbreviation for clear distinction makes sense.

The limitation to non-stochastic environments is a clear limitation of this work. This is acknowledged and stated to be left for future work in 4.1, but I think this limitation should already be acknowledged earlier in the paper.

I think releasing the code and full config files would certainly strengthen reporducibility and enhance the relevance for other researchers.

**Audience:**

Yes

**Audience Explanation:**

I think the paper has a potentially broad audience, as RL post-training is certainly an quite active research area. What I think is especially nice is that the method is relatively simple, and should be compatible with most standard GRPO pipelines (and maybe even beyond). Furthermore, the analysis of task space diversity -- although only briefly discussed in the paper -- should be ob interest to a broad number of researchers.

**Claims And Evidence:**

Yes

**Claims Explanation:**

I think the main claims (i.e. that EXIT improves self-improvement at inference and increases task diversity) are sufficiently supported by the empirical results on three different tasks. These advantages are apparent both for the initial accuracy, as well in the further gains due to the improvement steps.
I think also the experiments are reporducible due to the provided hyperparameter sets, and the ablation studies support sufficient generalization.

Some points that from my point of view still can be improved to enhance the significance of the results are listed below.

**Requested Changes:**

In general I think this is a quite nice paper presenting an interesting concept. As explained above, I think the main claims are decently supported, but I see following points for imporvement:

**My major points:**
* I think the statistical significance of the results is not entirely safe from debate, more concretely:
   * While some parts are noted with confidence intervals (or rather std in Table 1), most other plots do not show this information.
      * Especially for the Tool-Use environment the improvement for the K-steps seems to be rather small (in particular Fig. 3 middle + right), so hard to judge significance there
   * I am aware that the training runs are quite expensive on a individual basis, but only 3 runs per configuration seems a bit too small to guarantee significant results
* EXIT used $\Delta_{max}$ for normalization, but I couldn't find a concrete definition in the paper (apart from "is the largest possible improvement")
   * How can this term be evalauted / estimated for an arbitrary environment?
   * I think it should be explicitly provided for the three considered tasks.
* The paper claims that single-step training suffices and is more efficient, so I think this should be validated against another baseline with multi-step RL training (i.e. fixed K)

**Some smaller things:**
   * I think the "compute-equivalent" claim when using variance-based prioritization should be supported (e.g. some wall-clock times, overhead of sampling from and maintaining buffer, ...)
   * The difference of the diversity metrics considered in Table 2 vs. Figure 7 might need some more explanation and contextualization. For me it was a bit difficult to understand what is actually evaluated there, and how these two analysis do or no not relate.

---

### Review · Reviewer_qHPq · 2025-10-16

**Summary Of Contributions:**

The paper introduces *Exploratory Iteration (ExIt)*, a reinforcement learning strategy for training LLMs to perform multi-step self-improvement at inference time using only single-step training iterations. ExIt maintains a buffer of high-learnability partial histories and selectively samples them for further improvement or divergence to expand the task space dynamically. The authors tested math reasoning, multi-turn tool-use, and ML engineering tasks. ExIt improves both initial solution quality and iterative refinement ability, with diversity mechanisms further enhancing performance and task coverage.

**Additional Comments:**

None

**Audience:**

Yes

**Audience Explanation:**

The authors address self-improvement at inference time in LLMs, with Exit being a welcome addition in this LLM era.

**Claims And Evidence:**

Yes

**Claims Explanation:**

Yes, the authors have provided extensive empirical results.

**Requested Changes:**

- Reduce repetition in the text, particularly in sections 4.0 and 4.1 by consolidating overlapping explanations into a more concise presentation.
- Provide a clear analysis of the computational overhead involved in computing the diversity score (If I understood correctly it involves working in the embeding space) including its impact on training time and scalability.
- Elaborate how ExIt would perform with extremely large task spaces or very long multi-turn histories
- If possible provide an analysis of negative cases, such as scenarios where ExIt fails to improve performance or where diversity-seeking mechanisms hinder convergence

---

### Review · Reviewer_UHiV · 2025-11-14

**Summary Of Contributions:**

This paper introduces Exploratory Iteration (ExIt) that desighed to train LLMs to perform multi-step self-improvement at inference-time while only training on the most informative single-step iterations. The authors evaluate ExIt across three tasks: math, function call and ML engineering, which convincing the ablation results against several baselines like GRPO.

Strength:

1. This paper is solid. The presentation and writing is easy to follow with detailed proof process.

2. The investigated question that how to efficiently train models to perform multi-step self improvement is important.

3. The ablation experiments are conducting on three tasks: Math, Function call, ML engineering, which include a wide range of practical tasks. The authors include three kinds of models from LLaMA to Qwen and Deepseek.

4. The authors compare their methods with several baselines and outperform them on the three tasks.

**Audience:**

Yes

**Audience Explanation:**

The investigated problem is very novel and important. Because scaling test-time computation could obviously improve the performance and self-improvement is a key method of scaling test-time computation.

**Claims And Evidence:**

Yes

**Claims Explanation:**

The experimental results and claims have sufficient evidences in the main text.

**Requested Changes:**

Weakness:

1. Why are the buffer size of three tasks different? Is there any ablations to show the selected size is the best size?

2. I found three tasks use three different models. Why use this kind of settings? It would be better if there are more experimental results on the same models across different domains or tasks to show the generalization performance. Reporting different models on different tasks may be suspicion of selecting a suitable model for the task.

3. For math task, I couldn't find the accurate sub-task performance. In figure 4 and figure 8, it seems the improvement gains largely from the MATH500. For sub-tasks like AIME 2025, the improvements remain fair.

4. An average of 20 of Math related score falls behind existing methods (such as [1]). I recommend the authors to update the models to see if EXIT could further improve the performance.

[1] RLAD: Training LLMs to Discover Abstractions for Solving Reasoning Problems

---

### Author Response · Authors · 2026-02-09
**Response to reviewers and updates in the CRC**

We thank the reviewers for their generally favorable reception of our work. Many thanks to the Action Editor for recommending the acceptance of our paper.

We have updated the manuscript to address various comments made by reviewers. In particular, we have

- Added a "Future Work and Limitations" section to address
  - Reviewer UHiV and qHPq's suggestions to run additional experiments on more recent (and more powerful) open models, as well as on training domains with very long multi-turn histories (we cap the number of turns to 10 for the tool-use domain in our study).
  - Address Reviewer qHPq's feedback on the compute-equivalent claim by clarifying how the diversity-seeking (full) version of ExIt is only approximately compute equivalent, as the FLOPs are primarily concentrated training and inferencing the larger LLM, making it valuable to investigate cheaper metrics for diversity.
  - Discuss additional directions for extending ExIt that we believe are worth pursuing.

- Rephrased explanation of the diversity measures in Table 2 in Sec 5.5, based on FpLj's feedback.

Individual responses to reviewers' questions below:

### Reviewer UHiV

- Why are buffer sizes different?
  - We swept buffer size for each task and reported the best buffer size for each ablation (described in Appendix C).
  - All ablations for each task were performed on the same starting model. We made the deliberate choice to use the best open model available at the time for each task in order to focus on additional improvements that can be driven by our method.
  - Yes, much of the gains are driven by MATH500, likely because the MATH500 format is most similar to the training set used for the Qwen2.5-7B-Instruct model.
  - We agree it would be informative to run our method on more recent models that achieve higher math scores out of the box, though we leave this for future work. We made mention of this in the new "Limitations and future work" section.

### Reviewer qHPq

- We have added a clarification on the additional costs incurred by running the embedding model for full ExIt variant in the new "Limitations and future work" section.
- While our training settings limit the number of multi-turn histories to 10 (for tool-use), both the tool-use and MLE-bench settings make use of very long test-time multi-turn histories. We also highlight application to much longer histories during training in the "Limitations and future work" section.

### Reviewer FpLj

- We compute the maximum improvement as follows (see Appendix A.4): We set this value to the distance to the best possible value for the success metric used to compute Kaggle medal status and competition percentiles, e.g. for AUC this is 1.0, for a distance metric, this is 0.0.
- We believe the correct baseline here is the single-step training baseline, as our method only trains on a single-step at training time. Moreover, for many of the tasks in this study, a full K-step multi-step history would not fit in the models' context.
- We rephrased Sec 5.5 to clarify the diversity measurement used in Fig 7.

---

### Decision · Action_Editor_TBGN · 2026-01-14

**Recommendation:** Accept with minor revision

**Additional Comments:**

The reviewers did not find any significant flaws in this work. However, they all pointed out some minor weaknesses that have not yet been addressed by the authors. For example, the experimental results show only incremental improvement. Therefore, the final recommendation is Accept with minor revisions. The revised version should address all the weaknesses and questions raised by the reviewers.

**Audience:**

Yes

**Audience Explanation:**

All reviewers agreed that this work tackles a novel and important problem.

**Claims And Evidence:**

Yes

**Claims Explanation:**

Two reviewers have given Leaning Accept, and the other reviewer (UHiV), who has not yet submitted its final verdict, also expressed a positive stance in its previous feedback.